# High Efficiency Continuous-Variable Quantum Key Distribution Based on ATSC 3.0 LDPC Codes

**DOI:** 10.3390/e22101087

**Published:** 2020-09-27

**Authors:** Kun Zhang, Xue-Qin Jiang, Yan Feng, Runhe Qiu, Enjian Bai

**Affiliations:** School of Information Science and Technology, Donghua University, Shanghai 201620, China; 2181282@mail.dhu.edu.cn (K.Z.); 2151167@mail.dhu.edu.cn (Y.F.); qiurh@dhu.edu.cn (R.Q.); baiej@dhu.edu.cn (E.B.)

**Keywords:** continuous-variable quantum key distribution, reconciliation efficiency, low-density parity-check codes

## Abstract

Due to the rapid development of quantum computing technology, encryption systems based on computational complexity are facing serious threats. Based on the fundamental theorem of quantum mechanics, continuous-variable quantum key distribution (CVQKD) has the property of physical absolute security and can effectively overcome the dependence of the current encryption system on the computational complexity. In this paper, we construct the spatially coupled (SC)-low-density parity-check (LDPC) codes and quasi-cyclic (QC)-LDPC codes by adopting the parity-check matrices of LDPC codes in the Advanced Television Systems Committee (ATSC) 3.0 standard as base matrices and introduce these codes for information reconciliation in the CVQKD system in order to improve the performance of reconciliation efficiency, and then make further improvements to final secret key rate and transmission distance. Simulation results show that the proposed LDPC codes can achieve reconciliation efficiency of higher than 0.96. Moreover, we can obtain a high final secret key rate and a long transmission distance through using our proposed LDPC codes for information reconciliation.

## 1. Introduction

Quantum key distribution (QKD) [1,2,3] is one of the most pragmatic applications of quantum communication technologies. QKD enables two remote parties named Alice and Bob to extract a symmetrical string of secret keys via a quantum channel, even in the presence of an eavesdropper, and promises unconditional security in principle [4]. Currently, there are two types of protocol, including discrete variable (DV) QKD [5] and continuous-variable (CV) QKD [6,7,8].

In DVQKD, the polarization or phase of weak coherent states is used as the carrier of information and the quantum states received are measured by using a single-photon detector. In continuous-variable quantum key distribution (CVQKD), the amplitudes and phase quadratures of Gaussian-modulated coherent states are considered to be the carriers of information and the receiver utilizes homodyne or heterodyne detection techniques to measure the transmitted quantum states [9]. For a CVQKD system based on Gaussian-modulated coherent states, future-proof security against collective attacks has been provided. Furthermore, such detectors used in CVQKD systems are routinely deployed in classical optical communications; hence, the CVQKD system offers very good prospects for implementations based on mature telecom components [4].

Generally, the quantum communication link and the information postprocessing are two parts of the CVQKD system [10,11]. In the quantum communication link part, the quantum states are prepared on Alice’s side and then sent to Bob via a quantum channel. Then, Bob performs a measurement on quantum states by using a homodyne detector. In the information postprocessing part, both Alice’s side and Bob’s side process the information and get symmetrical secret keys through correcting errors. Technically, the reconciliation efficiency is one of major factors limiting the secret key rate and the transmission distance of a CVQKD system [12]. Therefore, a high reconciliation efficiency is necessary to realize a high secret key rate and long transmission distance. The ideal secret key rate $K_{ideal}$ is given by
(1)$$K_{ideal}=\beta I_{AB}-\chi_{BE},$$
where the reconciliation efficiency is measured by β=R/C(s), *R* is the adopted rate of low-density parity-check (LDPC) codes and C(s) is the channel capacity at signal-to-noise ratio (SNR) value *s*. *I*${}_{AB}$ denotes the mutual information between Alice and Bob. The maximum of the Holevo information leaked to Eve is denoted by $\chi_{BE}$. However, secret key rate *K* in Equation (Equation 1) is the maximum achievable value without considering reconciliation frame error rate (FER). Taking the reconciliation FER into account, the final secret key rate of the CVQKD system is given by [13,14]
(2)$$K_{final}=\left(1-\mathrm{FER}\right)\left(\beta I_{AB}-\chi_{BE}\right).$$

It can be clearly seen from Equation (Equation 2) that for a given FER, in order to obtain a high secret key rate, the reconciliation efficiency should be as large as possible. Therefore, researchers begin to discover more efficient error-correcting codes to improve reconciliation efficiency. An expansion operation is proposed in [15], which provides a way to use quasi-cyclic (QC)-LDPC codes in the 5G protocol for information reconciliation and achieves high efficiency above 0.926 at rate 22/68. In [13,16,17], repeat accumulate (RA) LDPC codes, multi-edge type (MET) LDPC codes and punctured LDPC codes are adopted in the information reconciliation of the CVQKD system respectively.

Generally, for a given LDPC code, low SNR is required to achieve high reconciliation efficiency β. Compared with the LDPC codes mentioned above, Advanced Television Systems Committee (ATSC) LDPC codes can require a lower SNR under the same FER and code rate *R*, which means that a lower channel capacity and a higher reconciliation efficiency can be obtained by applying ATSC LDPC codes to the CVQKD system. The ATSC 3.0 standard is a new international broadcasting standard formulated by ATSC in 2013. The ATSC 3.0 standard gives an encoding algorithm for LDPC codes and supports code rates in the range of 2/15 to 13/15, and it has capacity-approaching performance and low encoding/decoging complexity [18,19,20,21].

In this paper, we construct spatially coupled (SC)-LDPC codes and QC-LDPC codes based on ATSC LDPC codes. The structure of QC-LDPC codes is convenient for memory storage and has low encoding/decoging complexity, while the variable nodes at the boundary of SC-LDPC codes are more likely to be successfully decoded. Due to the special structures of QC-LDPC codes and SC-LDPC codes, we can further improve the reconciliation efficiency by constructing QC-LDPC codes and SC-LDPC codes based on ATSC LDPC codes and then employ these codes to the CVQKD system. The numerical results show that the proposed codes have higher reconciliation efficiency than the original codes, which can improve the performance of secret key rate and transmission distance.

This paper is organized as follows. In Section 2, we provide preliminaries on multidimensional reconciliation protocol and LDPC codes in ATSC 3.0. In Section 3, we introduce the proposed SC-LDPC codes and QC-LDPC codes based on ATSC LDPC codes for information reconciliation. Section 4 shows the performance of our proposed LDPC codes. Finally, the conclusions are drawn in Section 5.

## 2. Preliminaries

In this section, we first briefly present the multidimensional reconciliation protocol in the CVQKD system, and then introduce ATSC LDPC codes, which will be used to construct error-correcting codes for information reconciliation part.

### 2.1. Multidimensional Reconciliation Protocol

In this work, we adopt reverse reconciliation scheme due to its longer transmission distance. The schematic diagram of multidimensional reconciliation protocol is shown in Figure 1, which can be described in the following steps:Alice’s side prepares Gaussian sequence *X* and sends it to Bob through a quantum channel to get sequence *Y*. Then, Alice and Bob divide *X* and *Y* into groups; each group consists of *n* elements, where *n* denotes the dimension of multidimensional reconciliation. Technically, we mainly adopt 8-dimensional reconciliation in the CVQKD system since it outperforms the other dimensions in reconciliation (n=1,2,4) [22].Alice and Bob normalize *X* and *Y* into *x* and *y*, respectively, where *x* = $X/∥X∥$ and $y=Y/∥Y∥$.A binary sequence *u* is randomly generated, and then sequence *e* is generated by sequence *u* through LDPC encoding. In order to get sequence *e* used in the multidimensional reconciliation, *e* needs to be converted to a binary spherical sequence $e^{\prime }$ as follows:
(3)$$\left(e_{1}^{\prime },e_{2}^{\prime },\ldots ,e_{n}^{\prime }\right)=\left(\frac{{\left(-1\right)}^{e_{1}}}{\sqrt{n}},\frac{{\left(-1\right)}^{e_{2}}}{\sqrt{n}},\ldots ,\frac{{\left(-1\right)}^{e_{n}}}{\sqrt{n}}\right).$$Bob’s side begins to calculate rotation function $M\left(y,e^{\prime }\right)=\begin{array}{c} \sum_{i=1}^{n}\alpha_{i}\left(y,e^{\prime }\right)A_{i} \end{array}$ and syndrome S=H·e by sequence *y* and sequence *e*, where $\alpha_{i}\left(y,e^{\prime }\right)=\left(A_{i}y|e^{\prime }\right)$ and specific matrix $A_{i}$ can be found in [23]; *H* is the parity-check matrix of LDPC code. Subsequently, Bob transmits the rotation function and syndrome to Alice through a classical channel.Alice receives the rotation function and maps her Gaussian sequence *x* to *c*, i.e., $c=M(y,e^{\prime })x$. Then, the parity-check matrices of the LDPC code, syndrome and sum-product algorithm are utilized to recover sequence *c* into sequence $u^{\prime }$. Alice and Bob will share symmetrical secret key when the decoding is successful.

We can greatly improve the final secret key rate and extend the safety distance of quantum communication by combining LDPC codes and multidimensional reconciliation. From Equation (Equation 2), we can see that the higher β of LDPC codes, the greater the performance improvement of multidimensional reconciliation for the CVQKD system.

### 2.2. LDPC Codes in ATSC 3.0

The LDPC codes in the ATSC 3.0 standard adopt two different lengths, commonly referred to as short codes (16,200) and long codes (64,800). There are two different coding structures for each code length, i.e., irregular repeat accumulate (IRA) and MET. For each code rate, the coding structure is determined by its bit error rate (BER) and FER performance over binary phase-shift keying/quadrature phase-shift keying modulation. The IRA structure shows better performance at high code rates while the MET structure shows better performance at low code rates [24]. The structure applied for each code rate and code length is shown in Table 1.

As shown in Figure 2, the IRA-structured parity-check matrix *H*, whose size is M×N, is composed of the information part *A* of size M×K and the parity part *D* of size M×M. The *A* part has a $Q_{1}$-shift structure while *D* part has a dual-diagonal structure.

The parity-check matrix of the MET-structured LDPC code is shown in Figure 3, which is composed of six parts where [*A**D*] in the upper left corner of the parity-check matrix has the IRA structure shown in Figure 2, *Z* part is a zero matrix of size $M_{1}\times M_{2}$, *I* part is an identity matrix of size $M_{2}\times M_{2}$ and [*B**C*] is a $Q_{2}$-shift structure of size $M_{2}\times \left(K+M_{1}\right)$.

Note that the MET-structured LDPC code can be seen as a concatenation of the IRA-structured LDPC code and many single parity-check codes which can provide an efficient encoding algorithm. In addition, the identity matrix has only one element in every row. Therefore, parity bits corresponding to the part *I* can be obtained in parallel and encoding latency can be correspondingly decreased [25].

In next section, we introduce the construction methods of our proposed parity-check matrices, which are used to correct errors in the noise version of secret key so that Alice and Bob can share the symmetrical secret key.

## 3. Proposed LDPC Codes for Information Reconciliation

It can be easily seen from Equation (Equation 2) that for a given FER, a high reconciliation efficiency results in a good performance of final secret key rate. In order to further improve the performance of reconciliation efficiency, we first construct SC-LDPC codes based on ATSC LDPC codes introduced in Section 2.2, and then construct QC-LDPC codes by adopting the parity-check matrices of ATSC LDPC codes as the base matrices.

Since the SNR of the quantum channel can be very low, we need LDPC codes of low code rates (i.e., R≤1/3) to be applied in the CVQKD system in order to show excellent error-correcting performance even at low SNR. As shown in Table 2, we constructed SC-LDPC codes and QC-LDPC codes based on ATSC LDPC codes at the code rates of 2/15 and 5/15, and applied these codes to the information reconciliation part of the CVQKD system.

### 3.1. Construction Method of SC-LDPC Codes for Information Reconciliation

The structure of the SC-LDPC code can be regarded as a special case of the protograph-based LDPC code. It links multiple unrelated subprotographs into a coupling link through spatial coupling [26], which results in a flexible construction structure for the SC-LDPC code. The detailed steps of our proposed SC-LDPC code based on ATSC LDPC code are described as follows:For a given parity-check matrix of ATSC LDPC code, we first calculate $\alpha =gcd(M_{b},N_{b})$, where gcd(·) represents the greatest common divisor function; $M_{b}$ and $N_{b}$ denote the number of rows and columns in the parity-check matrix, respectively.The cutting line is determined by alternately moving $N_{b}/\alpha$ entries to the right and then $M_{b}/\alpha$ entries down. Subsequently, we cut the parity-check matrix into upper and lower parts along the cutting line. After obtaining the upper and lower parts, we can construct a new parity-check matrix by re-splicing two parts as shown in Figure 4.We can repeat the step 2 until the parity-check matrix of desired code length is generated as shown in Figure 5.

Note that the $M_{b}\left(n_{b}+1\right)\times N_{b}n_{b}$ parity-check matrix of new SC-LDPC code is constructed from a smaller base matrix of ATSC LDPC code of size $M_{b}\times N_{b}$ via repeatedly copying and pasting as shown in Figure 5. For a given positive integer of repetitions $n_{b}$, the code rate of proposed SC-LDPC code is given by
(4)$$\begin{array}{cc} R_{SC} & =\frac{N_{b}n_{b}-M_{b}\left(n_{b}+1\right)}{N_{b}n_{b}}\\ \; & =\frac{N_{b}n_{b}-M_{b}n_{b}}{N_{b}n_{b}}-\frac{M_{b}}{N_{b}n_{b}}\\ \; & =R_{ATSC}-\frac{M_{b}}{N_{b}n_{b}}, \end{array}$$
which faces an unavoidable rate loss of $M_{b}/N_{b}n_{b}$ compared with the code rate of base matrix. Only when the number of repetitions is large enough can the effect of rate loss be approximately ignored.

For a simple example, let $M_{b}=3$, $N_{b}=6$ and repetitions $n_{b}=2$. Then, the base matrix $H_{b}$ of size $M_{b}\times N_{b}$ is given by
(5)$$H_{b}=\left[\begin{array}{cccccc} 1 & 0 & 1 & 0 & 0 & 1\\ 1 & 0 & 0 & 1 & 1 & 0\\ 0 & 1 & 0 & 1 & 0 & 1 \end{array}\right].$$

According to the construction steps of SC-LDPC codes, we can draw a cutting line in the $H_{b}$ by alternately moving $N_{b}/\alpha =6/3=2$ entries to the right and $M_{b}/\alpha =3/3=1$ entry down as shown in Equation (Equation 6):(6)
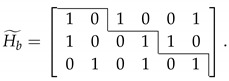


Subsequently, we can cut $\tilde{H_{b}}$ into upper and lower parts along the cutting line, and fill the missing entries of the two parts with 0s, which are given by
(7)$$H_{upper}=\left[\begin{array}{cccccc} 0 & 0 & 1 & 0 & 0 & 1\\ 0 & 0 & 0 & 0 & 1 & 0\\ 0 & 0 & 0 & 0 & 0 & 0 \end{array}\right],$$
(8)$$H_{lower}=\left[\begin{array}{cccccc} 1 & 0 & 0 & 0 & 0 & 0\\ 1 & 0 & 0 & 1 & 0 & 0\\ 0 & 1 & 0 & 1 & 0 & 1 \end{array}\right].$$

Accordingly, the SC-LDPC code based on base matrix $H_{b}$ will be generated by re-splicing the upper part and the lower part:(9)$$H_{SC}=\left[\begin{array}{cccccc} 1 & 0 & 0 & 0 & 0 & 0\\ 1 & 0 & 0 & 1 & 0 & 0\\ 0 & 1 & 0 & 1 & 0 & 1\\ 0 & 0 & 1 & 0 & 0 & 1\\ 0 & 0 & 0 & 0 & 1 & 0\\ 0 & 0 & 0 & 0 & 0 & 0 \end{array}\right].$$

Due to the $n_{b}=2$, we can get the new SC-LDPC code with code length of 12 by repeatedly copying and pasting Equation (Equation 9) as follows:(10)$$H_{SC}\left(n_{b}\right)=\left[\begin{array}{c} \begin{array}{cccccccccccc} 1 & 0 & 0 & 0 & 0 & 0 & 0 & 0 & 0 & 0 & 0 & 0\\ 1 & 0 & 0 & 1 & 0 & 0 & 0 & 0 & 0 & 0 & 0 & 0\\ 0 & 1 & 0 & 1 & 0 & 1 & 0 & 0 & 0 & 0 & 0 & 0\\ 0 & 0 & 1 & 0 & 0 & 1 & 1 & 0 & 0 & 0 & 0 & 0\\ 0 & 0 & 0 & 0 & 1 & 0 & 1 & 0 & 0 & 1 & 0 & 0\\ 0 & 0 & 0 & 0 & 0 & 0 & 0 & 1 & 0 & 1 & 0 & 1\\ 0 & 0 & 0 & 0 & 0 & 0 & 0 & 0 & 1 & 0 & 0 & 1\\ 0 & 0 & 0 & 0 & 0 & 0 & 0 & 0 & 0 & 0 & 1 & 0\\ 0 & 0 & 0 & 0 & 0 & 0 & 0 & 0 & 0 & 0 & 0 & 0 \end{array} \end{array}\right].$$

As can be seen from Equations (Equation 5) and (Equation 10), when the number of repetitions is two, the code length increases from 6 to 12 while the code rate reduces from 1/2 to 1/4, which results in inescapable rate loss of 1/4. However, we can increase the number of repetitions to lower the rate loss $M_{b}/N_{b}n_{b}$. For instance, the rate loss of new SC-LDPC code will be decreased to 3/(6×100)=1/200 if $n_{b}$ increases to 100, which is much smaller than the rate loss of 1/4 when $n_{b}=2$.

### 3.2. Construction Method of QC-LDPC Codes for Information Reconciliation

The parity-check matrix of the QC-LDPC code is composed of the zero matrices and the cyclic permutation matrices (CPMs). All submatrices are square matrices of the lifting size L×L [27]. Therefore, the QC-LDPC code is easy to implement by hardware and has low encoding/decoging complexity. Generally, we set parameter *p* as the number of cyclic shifts; the CPMs are matrices that shift the positions of 1s in the identity matrices to the right by *p* times. For example, if *p* = 1, the CPM can be expressed as:(11)$$H_{CPM}={\left[\begin{array}{cccccc} 0 & 1 & 0 & 0 & \cdots & 0\\ 0 & 0 & 1 & 0 & \cdots & 0\\ 0 & 0 & 0 & 1 & \cdots & 0\\ \vdots & \vdots & \vdots & \vdots & \ddots & \vdots \\ 0 & 0 & 0 & 0 & \cdots & 1\\ 1 & 0 & 0 & 0 & \cdots & 0 \end{array}\right]}_{L\times L}.$$

The structure of the parity-check matrix in ATSC LDPC code can be regarded as the special case of the base matrix for QC-LDPC code. We can construct QC-LDPC code based on ATSC LDPC code by replacing 0 s and 1 s in the base matrix with zero matrices and corresponding CPMs, respectively. The size of each submatrix is L×L. Furthermore, the *p* is set to a random number from 0 to L−1. For a given parity-check matrix of ATSC LDPC code, the structure of QC-LDPC code can expand the base matrix of size $M_{b}\times N_{b}$ to a size of $M_{b}L\times N_{b}L$. According to the cycle free features, the girths of the constructed QC-LDPC codes are no less than those of the original LDPC codes [28].

For convenience of presentation, we divide the parity-check matrix of MET-structured LDPC code into three parts, namely, an IRA structure which is embedded in the MET structure, a [*B*
*C*
*I*] part which is cascaded with the IRA structure and the all-zero part *Z*. Each element in the *Z* part will be replaced by the L×L all-zero matrix. The construction of QC-LDPC code based on IRA-structured LDPC code is shown in Figure 6.

Since the IRA structure is embedded in MET structure, the [*B*
*C*
*I*] part can be extended to QC-LDPC code in a same manner as shown in Figure 7.

For a simple example, we consider constructing the base matrix as shown in Equation (Equation 12),
(12)$$H_{b^{\prime }}=\left[\begin{array}{cccccc} 1 & 1 & 1 & 0 & 0 & 0\\ 1 & 1 & 0 & 1 & 0 & 0\\ 0 & 0 & 0 & 1 & 1 & 1 \end{array}\right],$$
into the form of a QC-LDPC code. Assume that the number of lifting size *L* is 3; the parameter *p* is set to a random integer from 0 to 2. Each 0 and 1 of the base matrix will be replaced by all-zero matrix and random CPM of size L×L, respectively. The corresponding CPMs are $H_{p=0}=\left[\begin{array}{ccc} 1 & 0 & 0\\ 0 & 1 & 0\\ 0 & 0 & 1 \end{array}\right]$, $H_{p=1}=\left[\begin{array}{ccc} 0 & 1 & 0\\ 0 & 0 & 1\\ 1 & 0 & 0 \end{array}\right]$ and $H_{p=2}=\left[\begin{array}{ccc} 0 & 0 & 1\\ 1 & 0 & 0\\ 0 & 1 & 0 \end{array}\right]$. Then, the base matrix can be expanded as follows:(13)$$H_{QC}={\left[\begin{array}{c} \begin{array}{cccccccccccccccccc} 1 & 0 & 0 & 0 & 1 & 0 & 1 & 0 & 0 & 0 & 0 & 0 & 0 & 0 & 0 & 0 & 0 & 0\\ 0 & 1 & 0 & 0 & 0 & 1 & 0 & 1 & 0 & 0 & 0 & 0 & 0 & 0 & 0 & 0 & 0 & 0\\ 0 & 0 & 1 & 1 & 0 & 0 & 0 & 0 & 1 & 0 & 0 & 0 & 0 & 0 & 0 & 0 & 0 & 0\\ 1 & 0 & 0 & 1 & 0 & 0 & 0 & 0 & 0 & 0 & 1 & 0 & 0 & 0 & 0 & 0 & 0 & 0\\ 0 & 1 & 0 & 0 & 1 & 0 & 0 & 0 & 0 & 0 & 0 & 1 & 0 & 0 & 0 & 0 & 0 & 0\\ 0 & 0 & 1 & 0 & 0 & 1 & 0 & 0 & 0 & 1 & 0 & 0 & 0 & 0 & 0 & 0 & 0 & 0\\ 0 & 0 & 0 & 0 & 0 & 0 & 0 & 0 & 0 & 0 & 0 & 1 & 0 & 0 & 1 & 1 & 0 & 0\\ 0 & 0 & 0 & 0 & 0 & 0 & 0 & 0 & 0 & 1 & 0 & 0 & 1 & 0 & 0 & 0 & 1 & 0\\ 0 & 0 & 0 & 0 & 0 & 0 & 0 & 0 & 0 & 0 & 1 & 0 & 0 & 1 & 0 & 0 & 0 & 1 \end{array} \end{array}\right]}_{9\times 18}.$$

It is easy to see that the girth of base matrix in Equation (Equation 12) is only four, which results in slowing down the decoding speed or even failing to decode successfully since it gets less information from the external cycle. However, as shown in Equation (Equation 13), we eliminate the limitation that the girth is only four by constructing QC-LDPC code based on Equation (Equation 12), which means that the constructed LDPC code has better performance than the original one. For a given FER, our proposed LDPC codes require lower SNR than original LDPC codes, which results that our proposed LDPC codes can achieve lower channel capacity c(s) and higher reconciliation efficiency β.

Technically, the error-correcting performance of our proposed code is related to its corresponding base matrix. In this work, we adopt the parity-check matrix of ATSC 3.0 standard LDPC code as the base matrix due to its excellent error-correcting performance, which plays a vital role in assisting two parties to obtain symmetrical secret keys in the information reconciliation process. Furthermore, our proposed LDPC codes based on ATSC LDPC codes can achieve a very high reconciliation efficiency under a very low FER, which will be verified in the next section.

## 4. Performance Analysis

In this section, we analyze the performance of proposed LDPC codes in terms of reconciliation efficiency β, FER, final secret key rate and transmission distance. In order to balance the loss of code rate and the increase in construction complexity, we construct SC-LDPC codes of length 6,480,000 based on ATSC LDPC codes by setting $n_{b}=100$. Moreover, we construct QC-LDPC codes of length 648,000 based on ATSC LDPC codes with lifting factor L=10. For decoding of LDPC codes, a sum-product algorithm with maximum iteration number 300 is adopted. Furthermore, *n* is set to 8.

As can be seen from Figure 8, the ATSC LDPC codes can achieve β values of more than 0.9. Furthermore, it can be clearly seen that the β values of our proposed LDPC codes are much higher than those of the ATSC LDPC codes under the same code rates. When R=2/15, the SC-LDPC codes enable β values to increase by 0.0102, 0.0207, 0.0254, 0.0391 and 0.0455 for FERs of 0.5, 0.2, 0.1, 0.01 and 0.001, respectively. When R=5/15, the SC-LDPC codes enable β values to increase by 0.0026, 0.0074, 0.0111, 0.0172 and 0.022 for FERs of 0.5, 0.2, 0.1, 0.01 and 0.001, respectively. Furthermore, the QC-LDPC codes raise β values by 0.0082, 0.0186, 0.023, 0.0357 and 0.0374 under the code rate of 2/15 for FERs of 0.5, 0.2, 0.1, 0.01 and 0.001, respectively. Moreover, the QC-LDPC codes increase β values by 0.006, 0.0106, 0.0141, 0.0206 and 0.0257 under the code rate of 5/15 for FERs of 0.5, 0.2, 0.1, 0.01 and 0.001, respectively. The reconciliation efficiency of our proposed codes can reach up to 0.96 for FER of 0.2.

According to Equation (Equation 2), it is obvious that for a given FER, the reconciliation efficiency should be as large as possible in order to obtain a high final secret key rate. Our proposed LDPC codes can achieve much higher β values than original ATSC LDPC codes under the same FERs, which indicates that our proposed LDPC codes can lead to higher final secret key rate and longer transmission distance.

As shown in Figure 9, we also compared the FER curves of ATSC LDPC codes and proposed LDPC codes under code rates of 2/15 and 5/15, respectively. It is easy to see that for a given FER, our proposed LDPC codes require lower SNRs than original ATSC LDPC codes under the same code rate, which results in smaller C(s) and higher β. Therefore, our proposed SC-LDPC codes and QC-LDPC codes will lead to higher reconciliation efficiency.

Figure 10 depicts the final secret key rate with respect to the transmission distance for the CVQKD system with different LDPC codes under FER of 0.001. The multidimensional reconciliation protocol in Figure 1 was used to obtain a high final secret key rate. Furthermore, the related parameters are as follows: excess noise ξ=0.012, attenuation factor of the quantum channel α=0.2 dB/km, efficiency of the homodyne detection η=0.6 and the electronic noise $\nu_{el}=0.055$. It is obvious that we can effectively increase the transmission distance by improving reconciliation efficiency. The performance of multidimensional reconciliation protocol combined with our proposed LDPC codes is significantly better compared to the original ATSC LDPC codes. This comes from the fact that original ATSC LDPC codes achieve lower reconciliation efficiency values, which are less than 0.91 when the FER is 0.001, while our proposed LDPC codes can achieve reconciliation efficiency values of more than 0.95 when the FER is 0.001.

## 5. Conclusions

In conclusion, we combined a multidimensional reconciliation protocol with our proposed LDPC codes based on ATSC LDPC codes. Generally, for a CVQKD system, we needed to improve the reconciliation efficiency in order to achieve higher performances of secret key rate and transmission distance. In this paper, our aim was to maximize reconciliation efficiency β for a given FER. The simulation results showed that ATSC LDPC codes can achieve high reconciliation efficiency, which resulted in good performance of secret key rate and transmission distance. Furthermore, the SC-LDPC codes and QC-LDPC codes that we constructed can further improve the performance of reconciliation efficiency, which resulted in better performance of final secret key rate and longer transmission distance. The efficiency of our proposed codes can reach up to 0.96.

## Figures and Tables

**Figure 1 entropy-22-01087-f001:**
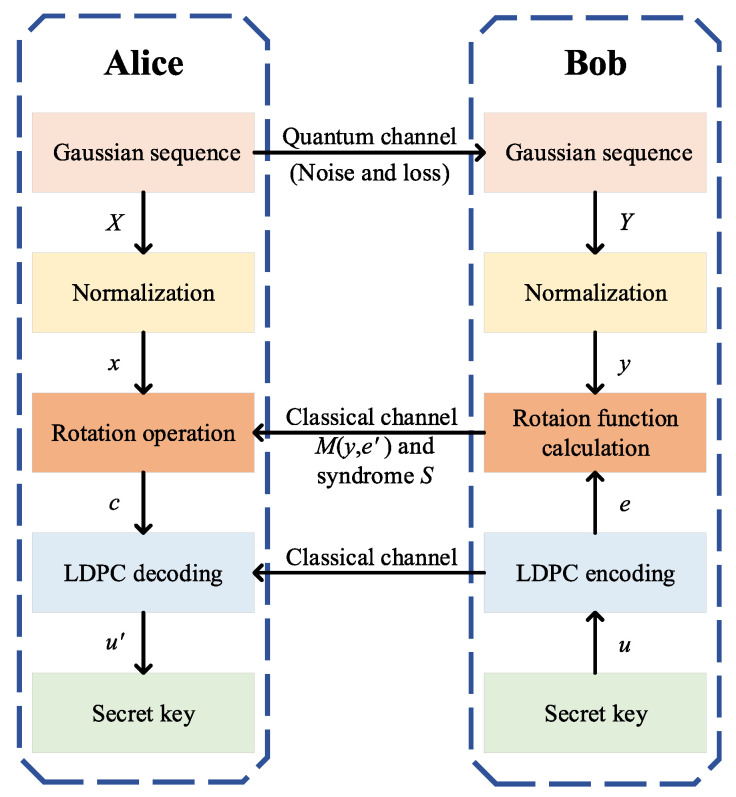
Schematic diagram of multidimensional reconciliation protocol. *X* and *Y* denote two corresponding Gaussian sequences. *x* and *y* represent the normalized results of *X* and *Y*. $M(y,e^{\prime })$ and *S* are the rotation function and syndrome, respectively. *u* is a random binary codeword and *e* is the LDPC encoding result of *u*. *c* denotes the result of rotation operation and $u^{\prime }$ represents the LDPC decoding result of *c*.

**Figure 2 entropy-22-01087-f002:**
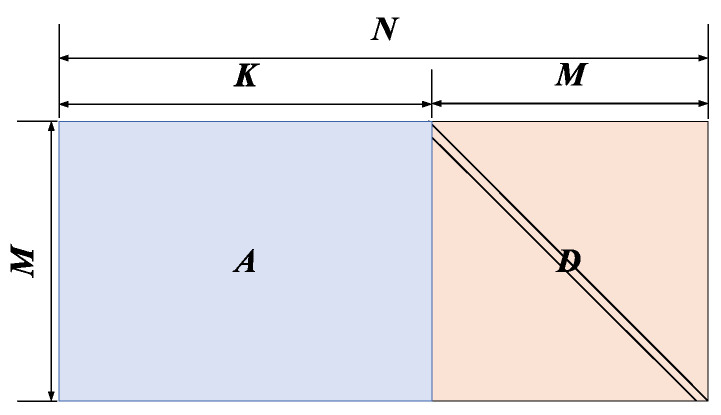
The parity-check matrix *H* of irregular repeat accumulate (IRA)-structured LDPC code.

**Figure 3 entropy-22-01087-f003:**
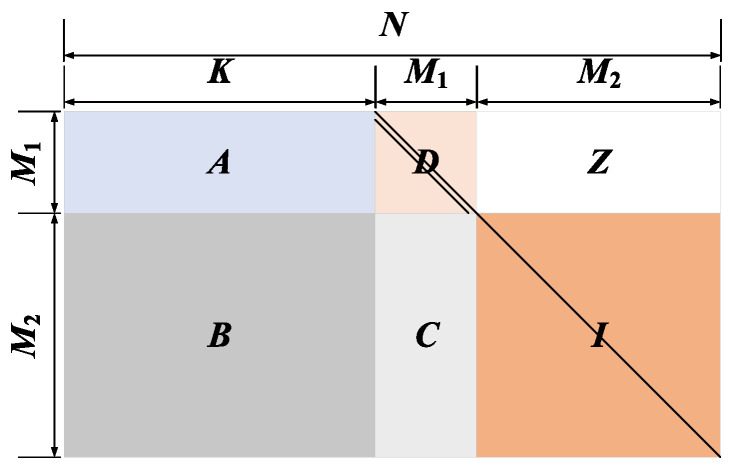
The parity-check matrix *H* of multi-edge type (MET)-structured LDPC code.

**Figure 4 entropy-22-01087-f004:**
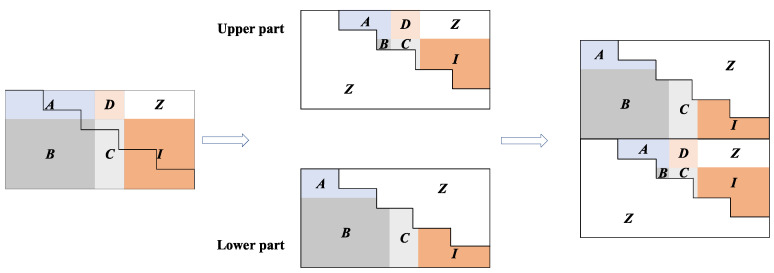
The construction of SC-LDPC code based on ATSC LDPC code.

**Figure 5 entropy-22-01087-f005:**
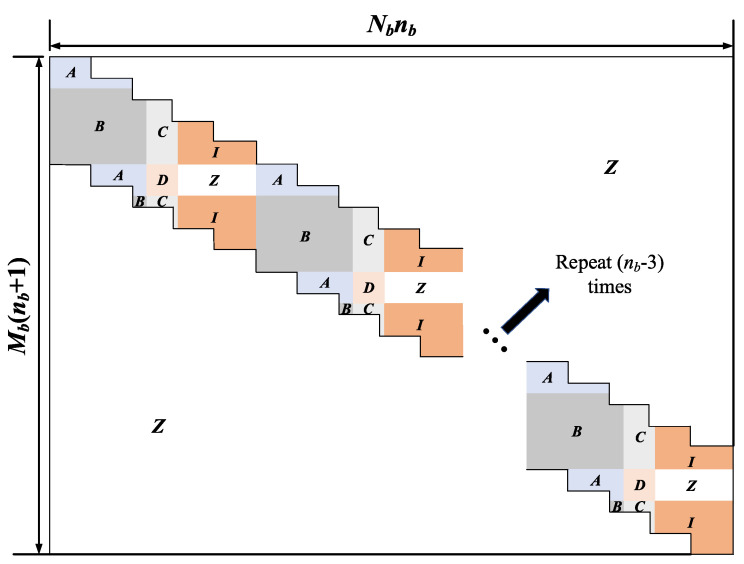
New SC-LDPC code with repetitions $n_{b}$.

**Figure 6 entropy-22-01087-f006:**
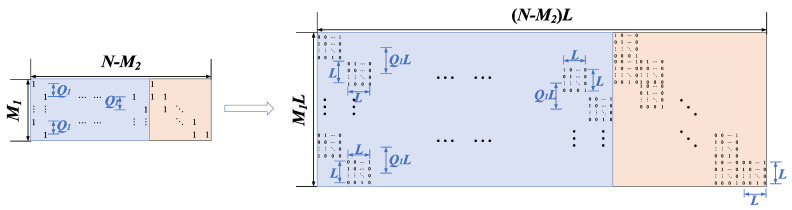
The construction of quasi-cyclic (QC)-LDPC code based on IRA-structured LDPC code.

**Figure 7 entropy-22-01087-f007:**
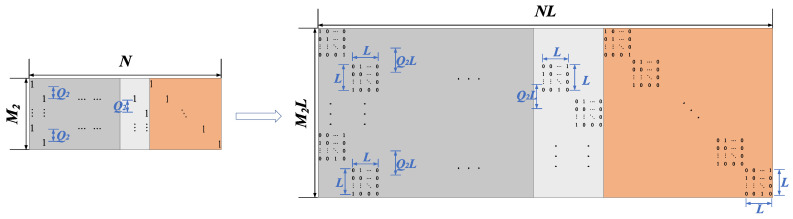
The construction of QC-LDPC code based on [*B*
*C*
*I*] part.

**Figure 8 entropy-22-01087-f008:**
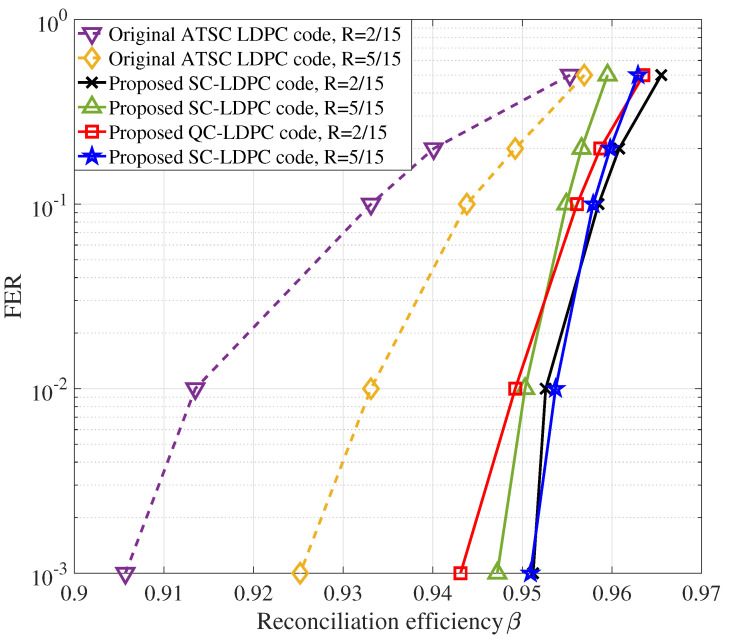
Reconciliation efficiency under different frame error rates (FERs).

**Figure 9 entropy-22-01087-f009:**
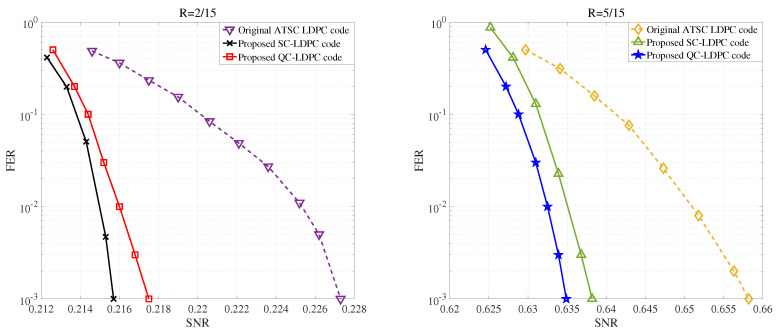
FER values corresponding to ATSC LDPC codes and proposed spatially coupled (SC)-LDPC codes, QC-LDPC codes for different SNRs under the (**Left**) code rates R=2/15 and (**Right**) R=5/15, respectively.

**Figure 10 entropy-22-01087-f010:**
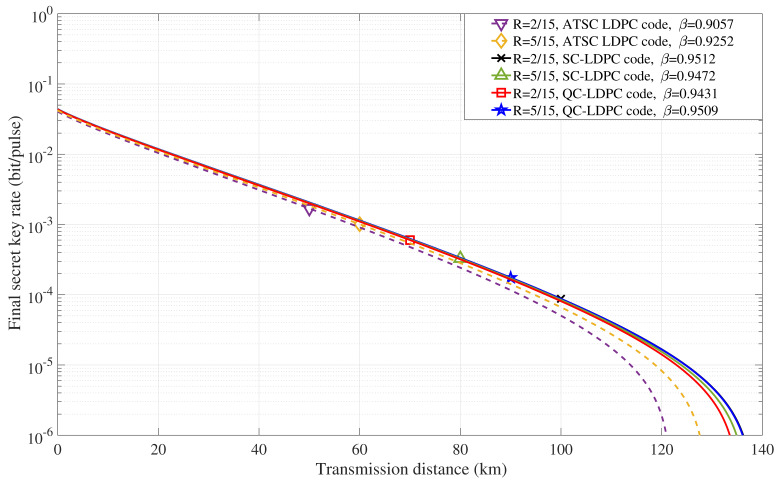
Final secret key rate as a function of transmission distance.

**Table 1 entropy-22-01087-t001:** The coding structures corresponding to code length and code rates.

Code Rate	Coding Structures of 64,800	Coding Structures of 16,200
2/15	MET	MET
3/15	MET	MET
4/15	MET	MET
5/15	MET	MET
6/15	IRA	IRA
7/15	MET	IRA
8/15	IRA	IRA
9/15	IRA	IRA
10/15	IRA	IRA
11/15	IRA	IRA
12/15	IRA	IRA
13/15	IRA	IRA

**Table 2 entropy-22-01087-t002:** The coding structures corresponding to different code lengths and code rates.

Code Rate	Original Structures	Proposed Structures of 648,000	Proposed Structures of 6,480,000
2/15	MET	QC	SC
5/15	MET	QC	SC

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
