# Peer review of "High Efficiency Continuous-Variable Quantum Key Distribution Based on ATSC 3.0 LDPC Codes"

_entropy, 2020, doi:10.3390/e22101087_

Round 1

Reviewer 1 Report

In the manuscript entitled “High Efficiency Continuous-Variable Quantum Key Distribution Based on ATSC 3.0 LDPC Codes”, Zhang et al. improved the original Advanced Television Systems Committee (ATSC) low-density parity-check (LDPC) code by modifying part of the code with more delicate spatially-coupled (SC) and quasi-cyclic (QC) parity check designs. Based on the improvements, they realize a high reconciliation efficiency of up to 0.96.

The manuscript is well-written and the results are useful for the real implementation of CV-QKD. I suggest its publication on Entropy. Below are some comments:

  1. The authors may clarify why use IRA encoding for the lower code rate 6/15, while use MET encoding for the higher code rate 7/15.
  2. Why do the authors only revise the coding structure of the two cases, 2/15, 5/15? Can we extend QC and SC encoding methods to other code rate cases? If not, what is the reason of that?
  3. The authors choose to group n=8 elements together to realize the LDPC encoding. Can the performance become better if we encode more elements in a group?
  4. Can the method be applied to discrete-variable QKD? Note that this information reconciliation part is very similar in the two cases, see e.g., [RMP 92, 025002 (2020)].

Reviewer 2 Report

The paper examines a rather complex and important issue in the field of CV QKD with Gaussian modulation, related to the reconciliation procedures and security enhancement. The work sets out the mathematical calculations in sufficient detail and provides illustrative material that allows one to evaluate the performance of the system. However, there are several complaints about the text that clarified below:

- In Introduction section, there are flaws related to the description of the QKD protocols types. Firstly, one might get the impression that DV protocols exclusively use polarization coding. Secondly, it is not clear why a source in a DV system has a "low successful probability of generating single-photon signal." DV protocols can be described in terms of weak coherent states, which require a laser source and an attenuator to generate. The states generation stage in CV QKD protocols with the so-called discrete modulation can be described in a same way. As a result of the above, the "limitations" that are removed by the utilized CV QKD protocol are not clear.

- Clarification regarding the protocol used is required. I understand that it is the Gaussian modulation protocol, but it should be noted. Also of note is the attack against which security is being assessed. This is not so important in the context of this article, but it is required for completeness.
